# Systematic Evaluation of How Indicators of Inequity and Disadvantage Are Measured and Reported in Population Health Evidence Syntheses

**DOI:** 10.3390/ijerph22060851

**Published:** 2025-05-29

**Authors:** Christopher J. Gidlow, Aman S. Mankoo, Kate Jolly, Ameeta Retzer

**Affiliations:** 1School of Medicine, Keele University, University Road, Newcastle under Lyme ST5 5BG, UK; 2Research and Innovation Department, Midlands Partnership University NHS Foundation Trust, St Georges Hospital, Corporation Street, Stafford ST16 3AG, UK; 3Centre for Health and Development (CHAD), University of Staffordshire, Leek Road, Stoke-on-Trent ST4 4DF, UK; a_mankoo@outlook.com; 4Institute of Applied Health Research, Murray Learning Centre, University of Birmingham, Birmingham B15 2TT, UK; c.b.jolly@bham.ac.uk (K.J.); a.retzer@bham.ac.uk (A.R.); 5National Institute for Health and Care Research (NIHR) West Midlands Applied Research Collaboration (ARC), Birmingham B15 2TT, UK; 6Centre for Patient Reported Outcomes Research, Institute of Applied Health Research, Murray Learning Centre, University of Birmingham, Birmingham B15 2TT, UK; 7NIHR Birmingham Biomedical Research Centre, Birmingham B15 2TT, UK

**Keywords:** population health, health inequality, health inequity, evidence synthesis, methodological review

## Abstract

We present a systematic evaluation of population health reviews from the Cochrane Database (January 2013–February 2023) to evaluate how indicators of inequity or disadvantage are considered and reported in population health evidence syntheses. Descriptive analyses explored a representation of reviews across health-determinant categories (primary and secondary categories), summarised equity-focused reviews, and examined proportions and types of reviews that planned/completed a subgroup analysis using ≥1 indicators from the PROGRESS-Plus framework. Of 363 reviews included, a minority focused on interventions targeting wider determinants of health (n = 83, 22.9% as primary category), with a predominance related to individual lifestyle factors (n = 155, 42.7%) or health care services intervention (n = 97, 26.7%). An explicit equity focus was evident in 21 (5.8%) reviews that used PROGRESS/PROGRESS-Plus, and 28 (7.7%) targeting vulnerable groups. Almost half (n = 165, 45.6%) planned a subgroup analysis by ≥1 PROGRESS-Plus indicator, which was completed in 63 reviews (38.2% of 165). The non-completion of planned subgroup analyses was attributed to insufficient data (too few primary studies, data not reported by subgroups). Among the 165 reviews planning a subgroup analysis, age was the most cited indicator (n = 91, 55.2%), followed by gender/sex (n = 67, 40.6%), place (n = 47, 28.5%) and socio-economic status (n = 37, 22.4%). This study highlighted missed opportunities for learning about the impacts of health equity in population health evidence syntheses due to insufficient data. We recommend routine use of PROGRESS-Plus and greater consistency in socio-economic proxies (occupation, education, income, disadvantage measures) to facilitate meta-analyses and subgroup analyses.

## 1. Introduction

Understanding and addressing the differential distribution of health outcomes and intervention effects across population groups are a public health priority [1], recognised by public health researchers, practitioners and policy makers [2,3].

The consideration of health inequity is central to the concept of population health, notwithstanding the debate regarding definitions [4,5]. Population health is determined by the wider environmental context, and different population groups vary in exposure and vulnerability to health risks, thus creating inequities in health [6]. Population health interventions can, therefore, tackle the wider determinants, or target vulnerable populations (e.g., low-income groups, less educated, homeless) who have a greater risk of poor health given their physical, economic, and social circumstance [1,7]. This aligns with Marmot’s principle of proportionate universalism, of reducing health inequity through providing support proportionate to need, whilst also addressing the wider determinants [8].

Health *inequalities* are the observed differences in outcomes across individuals and groups defined on the basis of socio-economic factors (e.g., income, education), geography, individual characteristics (e.g., age, sex, ethnicity), and other social factors (e.g., homelessness [9]). These also serve as an indirect means of evaluating health *inequity* or *disparity* [10]: when differences in health are considered unfair and avoidable because they result from some kind of injustice [11]. Some health differences are unavoidable, such as those linked to genetic predisposition or age [2,9]. But public health policy to reduce population health differences are contingent on them being avoidable (inequitable). Therefore, consistent with a public health remit of reducing avoidable differences within the population’s health, and with terminology used in measurement frameworks [12,13] and guidance [2,12,14], this study considers the measurement and reporting of equity/inequity.

Evidence syntheses are the primary sources of evidence used to inform public health practice and policy [15]. By combining data from large numbers of subgroups across diverse populations and settings, such syntheses should facilitate explorations of equity [16]. Yet, meta-analyses are often undermined by heterogeneity of indicators to examine equity [17,18]. The last 20 years have seen a concerted effort in this area [19]. A 2003 framework of (in)equity indicators, PROGRESS [20], was later augmented to PROGRESS-Plus: Place of residence, Race/ethnicity, Occupation, Gender/sex, Religion, Education, Socioeconomic status, Social capital, and personal characteristics [21]. Subsequent guidance for systematic reviewers is offered in the 2012 PRISMA-Equity (or PRISMA-E) extension [12,14] and Cochrane handbook [2]. Yet there is scope for varied practice: which population characteristic to use and the potential conceptual overlap between them (e.g., place can be a proxy for various area-level characteristics), which indicators best represent a given characteristic (e.g., educational attainment or years completed) and how to categorise them (e.g., low/high; primary/secondary/tertiary education). The relevance and measurement of characteristics might also vary with geographical, social or cultural context (e.g., low–middle- vs. high-income countries).

Others examining equity reporting in evidence syntheses have confirmed inconsistency in how inequality/inequity is reported in health inequality/inequity-focused reviews. Hollands et al. [22] found many reviews that used or intended to use PROGRESS-Plus (through targeted searches). Others have observed varied use of PROGRESS/PROGRESS/Plus and PRISMA-E, and variation in the extent of subgroup analyses [23] and how checklists could or should be applied [22]. Welch et al. [24] reviewed 158 methodological studies that examined how systematic reviews assessed health equity. Most used a descriptive assessment of equity reporting and analysis (140, 88.6%), with 58 studies assessing whether reviewers conducted subgroup analysis using one or more PROGRESS-Plus characteristics.

The present study builds on these reviews, evaluating population health evidence syntheses, where health equity *should* be a routine consideration. Our aims were to: (i) explore the representation of population health reviews across health-determinant categories; (ii) describe those with an equity focus; (iii) examine the proportions and types of reviews that planned/completed subgroup analyses using ≥1 indicators from the PROGRESS-Plus framework.

## 2. Materials and Methods

### 2.1. Design and Search

This was a systematic evaluation of a cohort of Cochrane reviews of population health research [25]. It took the form of an overview of reviews [26] following a modified PRISMA reporting format (Appendix A). The data source was the complete Cochrane Database of Systematic Reviews from 1 January 2013 to 19 February 2023 (n = 5953). The full protocol is available at Research Registry (reviewregistry1717).

### 2.2. Inclusion and Exclusion Criteria

Inclusion/exclusion criteria were refined during an initial calibration phase. We aimed to capture evidence syntheses relevant to population health, with potential for sub-analysis (planned and/or completed), or that targeted ‘vulnerable’ populations. This was to ensure that reviews not specifying an equity focus (a noted weakness [12]), but that could or should have, were not missed. Potentially eligible types of syntheses included reviews of non-clinical interventions (e.g., population health, behavioural and educational interventions, mass media campaigns); reviews of reviews; rapid reviews of population health interventions; reviews of prognostic and non-clinical prototype public health studies.

Reasons for exclusion are detailed in Appendix A. Briefly, we excluded reviews of: diagnostic accuracy studies; qualitative studies; clinical intervention studies; clinical populations; other specific populations (e.g., women with multiple births, people undertaking cosmetic procedures); only low- and middle-income countries (LMIC) to avoid additional complexity through differences in the wider economic, geographical and health context between high-income countries (HIC) and LMICs [2], and related differences in equity measures [27]; health service design, organisation of care and healthcare professionals’ practice; and individual-focused intervention studies. We also excluded scoping reviews as they did not include quantitative syntheses with potential for subgroup analyses, and methodological reviews focused on the methods or processes of research (e.g., participant recruitment, retention or randomisation, or different statistical methods).

### 2.3. Screening

Following a calibration phase using 180 consecutive titles and abstracts (by CG/AR), finalised criteria were used to screen titles/abstracts of 5953 Cochrane Reviews (by CG), with independent assessment (by AM) of a 10% sample selected using an MS Excel random number formula (target of 90% agreement). Full texts of potentially eligible reviews were screened, with independent verification (by AM, 10% random sample, target of 90% agreement). Disagreements were resolved through discussion.

### 2.4. Outcome Selection

Key outcomes included: the mention of inequity, inequality or social patterning in the introduction/methods; the explicit use of PROGRESS/PROGRESS-Plus checklists; targeting a vulnerable population group; explicit planning and/or completing subgroup analyses using PROGRESS-Plus indicator types; indicators used; how they were categorised.

### 2.5. Data Extraction and Analysis

All data were extracted by CG using an MS Excel data extraction form (Appendix A) with independent verification by AM (10% random sample). All disagreements were resolved through discussion. Reviews were not quality appraised given the lack of relevant appraisal tools and limited guidance available for methodological reviews [28]. The analysis was descriptive: mapping reviews to the appropriate Dahlgren and Whitehead health-determinant categories [29], assigning primary and secondary categories if more than one was applicable; describing if/how equity was examined (Appendix A).

## 3. Results

### 3.1. Results of Screening

The screening of 5953 review title/abstracts identified 396 full texts that were assessed for eligibility (Figure 1). Thirty-three full texts were excluded, leaving a final sample of 363 reviews (Appendix A). The most common reasons for exclusion were study populations having existing conditions (n = 12) or including only studies from LMIC countries (n = 11) (Appendix A).

### 3.2. Characteristics of Reviews

When mapped to the determinants of health categories, the largest proportions of reviews mapped to individual lifestyle factors (n = 155) or health care services (n = 97) as a primary category, followed by education (n = 30), other (n = 24) and work environment (n = 22). Less than one-quarter of reviews aligned with wider determinants (n = 83 as primary category), reflecting physical living environment (e.g., housing, water sanitation) and broader socio-economic conditions (e.g., general socio-economic conditions, unemployment) or the social environment (social and community networks) (Figure 2; Appendix A).

### 3.3. Measurement and Reporting of Health Inequities or Inequalities

Half the reviews referred to inequalities, inequities or social patterning in the introduction/methods (n = 181, 49.9%). Twenty-eight (7.7%) focused on interventions targeting vulnerable populations, most commonly people with experience of abuse (n = 6), caregivers (n = 4) or people working in environments that expose them to risk (n = 3) (Table 1).

Twenty-one (5.8%) of the 363 reviews used PROGRESS [n = 7, 1.9%] [30,31,32,33,34,35,36] or PROGRESS-Plus [n = 14, 3.9%] [37,38,39,40,41,42,43,44,45,46,47,48,49,50] to extract data and consider equity impacts and disadvantage (Table 2). All were published after the 2012 PRISMA-E extension was introduced [12].

**Figure 1 ijerph-22-00851-f001:**
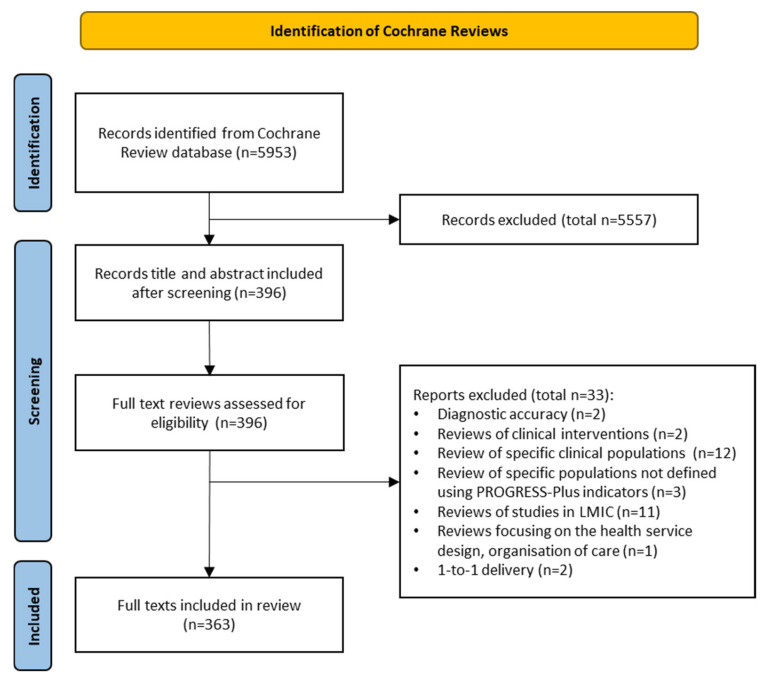
PRISMA flow diagram of study selection (adapted from [39]).

**Figure 2 ijerph-22-00851-f002:**
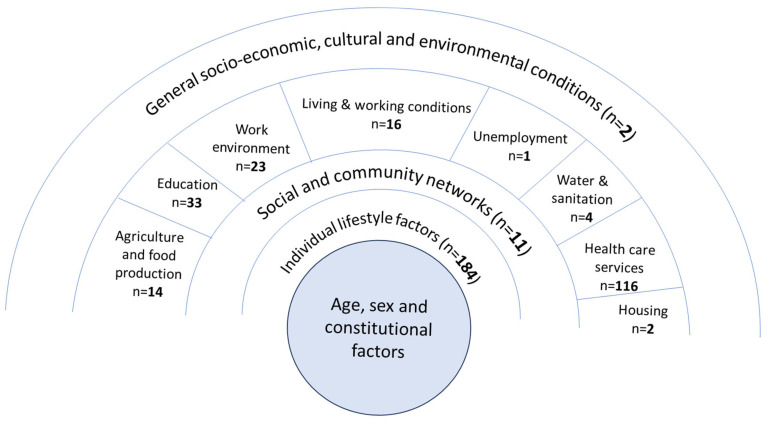
Number of reviews mapped to Dahlgren and Whitehead determinants of health categories (numbers reflect alignment as primary or secondary categories; therefore, the total number (n = 4354) exceeds the number of reviews included (n = 363); see Appendix A.

### 3.4. Use of PROGRESS/PROGRESS-Plus

Table 2 summarises the 21 reviews that used PROGRESS or PROGRESS-Plus. All mentioned inequity/inequalities in the introduction/methods, and one focused on a vulnerable group (young people experiencing homelessness [40]). Two reviews did not plan a subgroup analysis by any PROGRESS indicators despite the equity focus [38,46]. A review of psychosocial support for smoking cessation in pregnancy completed most analyses for three PROGRESS-Plus indicators, with a narrative synthesis for others [39].

Eleven reviews reported a narrative synthesis around equity, some with accompanying tables or appendices detailing equity considerations in each study. Of these, six also completed subgroup analyses for some/all of the intended indicators [30,34,36,47,49,50], whereas five planned but did not complete analysis [32,40,42,48,51]. Five reviews included less detailed narratives around equity. Key points were covered, but data were not presented by PROGRESS indicators, often due to a lack of information in primary studies [35,37,41,43,45]. Two reviews, both focused on taxation to improve the health content of food, offered limited consideration of equity impacts [31,33].

Of the 21 reviews that used PROGRESS/PROGRESS-Plus, the predominant determinant of health categories were agriculture and food production (n = 9) and individual lifestyle factors (n = 9) (Table 2).

### 3.5. Subgroup Analysis by PROGRESS-Plus Indicators

Less than half of the 363 reviews (n = 165, 45.7%) planned a subgroup analysis using one or more equity indicators, of which 63 (38.2% of 165, 17.4% of 363) completed analyses. Of the 181 reviews that mentioned inequality, inequity or social patterning in the introduction/methods, 110 (60.8%) planned a subgroup analysis, 71 (39.2%) did not.

Figure 3 illustrates the relative numbers of reviews aligned with each health-determinant category (brown bubbles—wider determinants; blue—other categories) and that planned subgroup analysis. Although relatively few focused on the wider health-determinants, within-category proportions that planned a sub-group analysis were highest for these categories (≥60%, work environment, unemployment, general socio-economic, agriculture and food production, living and working conditions and education). Conversely, for reviews mapped to healthcare services and individual lifestyle factors, the proportions planning a subgroup analysis were lower (≤40%), despite a greater number of reviews.

Figure 4 shows the relative numbers of reviews that planned and completed a subgroup analysis, by health-determinant category. Small numbers make the percentages less meaningful, but for categories with 10 or more reviews, the completion rates ranged from 4.3% (water and sanitation) to over 25% (living and working conditions, agriculture and food production). For the most common categories of individual lifestyle factors and healthcare service categories, there were low rates of planned (40.2% and 37.9%) and completed (20.7% and 15.5%) subgroup analyses.

One hundred of the 102 (98.0%) reviews that planned, but did not complete subgroup analyses, cited insufficient data (too few primary studies overall or that reported outcomes by subgroups, or insufficient heterogeneity); two did not specify.

The indicators most intended for/used in subgroup analyses were age, followed by gender/sex, place, socio-economic status (SES) and race/ethnicity (Table 3; Appendix A). There was varied practice. Age categories varied with context and target population, including specific and broad age groups, life stage or school year/stage. Gender/sex groups included a mix of gendered labels (boys/girls, men/women, mothers/fathers) and sex at birth (male/female). Place was mostly country level income (low–middle/high income) or urban/rural location. SES included a range of indicators of disadvantage/social disadvantage/deprivation, income and generic or unspecified proxies. Race/ethnicity groups reflected foci on both ethnicity and race and often compared majority/minority groups, indigenous and non-indigenous groups or groups based on disease risk. In many cases, there was a lack of specificity. Planned and completed subgroup analyses using other equity indicators were uncommon, with few examples of the ‘Plus’ indicators being considered, aside from age (Table 3).

## 4. Discussion

### 4.1. Principal Findings

This systematic evaluation of inequity considerations in population health evidence syntheses confirmed varied practices and limitations, which ultimately limit the evidence base to address health inequity. Although half of the 363 population health reviews (49.9.%) referenced inequities, inequalities or social patterning in their background/methods, few demonstrated a specific equity focus through citing PROGRESS/PROGRESS-Plus (n = 21, 5.8%), in line with the PRISMA-E [2,12], or through targeting vulnerable populations (n = 28, 7.7%).

Compared with other methodological reviews that focused on wider determinants [52] or delimited to health-equity-focused reviews [22,23,24,53], our population health focus was more inclusive. It presents a more critical picture: just 5.8% of the 363 reviews that *should* or *could* have used PROGRESS-Plus for data extraction (as a minimum) did so. This accords with Welch et al. [24] who found that, even among methodological studies of health equity assessments in systematic reviews, just 18 out 158 (11.4%) studies explicitly cited PROGRESS-Plus. A comparison with equivalent proportions reported by Hollands et al. [22] is less useful given their purposive article selection to capture the breadth of approaches (rather than comparing proportions using different approaches). In the present study, it is possible that some review authors, particularly those examining interventions not targeting the wider determinants (77% of included reviews), might not have treated their topic as *population health* and, therefore, not judged equity as a necessary consideration. However, regardless of the authors’ explicit focus, we identified an implicit link to population health and, often, a missed opportunity for equity consideration. Those focused on vulnerable groups explored equity impact through studying populations at greater risk of poor health as a result of unsupportive physical, economic or social circumstances [7]. This might explain why subgroup analysis was planned in only 11 of 28 reviews (completed in just one, by age and gender [54]). There are other examples when certain subgroup analyses would be unnecessary (e.g., by gender if interventions target women; by age if interventions target specific age groups; by occupation if targeting children/older adults/unemployed). These caveats notwithstanding, 39% of reviews that mentioned inequality, inequity or social patterning in the introduction or methods, did not plan any subgroup analysis to examine equity impact. This indicates that equity consideration and reporting are often not routine practice.

We focused on use of subgroup analysis as it allows reviewers to inspect broad patterns of inequity, the limitations aside (e.g., indicators considered in isolation; inability to infer causality [23]). Almost half (45.7%) of the 363 reviews planned subgroup analyses and 17.4% were able to complete some, despite only 5.8% citing PROGRESS/PROGRESS-Plus. Authors of included reviews noted insufficient data to allow analysis through primary studies being too few or not reporting data by subgroups, although the rate of subgroup analyses by one or more PROGRESS-Plus indicator was higher than the 8% (of 262 reviews, across 58 methodological studies) observed by Welch et al. [24].

This deficit in equity reporting among primary studies was reported in a recent assessment of 200 ‘equity-relevant’ studies. Karran et al. [53] found that most studies reported age, sex/gender and education (92%, 78% and 65%, respectively), and approximately half reported other socio-economic proxies (49%), race/ethnicity (45%) and social capital (44%). Yet their analysis identified an overall inadequacy and inconsistency of equity reporting which, they concluded, would likely undermine opportunities to pool data. Indeed, they illustrated that many primary studies fell short of the recommended minimum requirements for sample description by age, gender, ethnicity and a socio-economic measure [55], similar to limitations noted elsewhere [56]. We similarly found that age and gender/sex subgroups were the most used in subgroup analysis, followed by place, race/ethnicity and SES (Table 3). The low rate of planned subgroup analysis by SES (22.4%) and completed analysis in just 4.2% was particularly striking and lower than observed elsewhere [24]. This low rate, in general and relative to other PROGRESS-Plus indicators in the present review, perhaps reflects our broader population health focus, but again shows a deficit in practice.

The relative sparsity of reviews relating to the wider determinants of health speaks to a continued need for further studies and reviews, particularly given that upstream conditions are regarded the key drivers of health and equity [1]. Within this sample of 363 population health evidence syntheses, few aligned with the wider determinants, but were considered ‘population health’ given the potential to affect the social environment (e.g., social support, group-based programmes, family programmes). This under-representation was similar to that observed in a rapid review of population health reviews by Retzer et al. [57]. It reflects a general and longstanding deficit in evidence of how interventions on social determinants impact health inequity [52]. Practical challenges to developing this evidence include the feasibility of intervening in wider environmental conditions. This might require substantial investment (e.g., physical infrastructure), political buy-in (e.g., policy change) or take many years (e.g., housing development), in addition to the complexities of measuring the health effects.

### 4.2. Strengths and Limitations

*Strengths*. The strengths of this systematic evaluation included a population health focus that considered a range of health-determinant categories. The use of the Cochrane Review library as a pragmatic approach offered some advantages regarding quality assurance [58,59] and consistency in the reporting of subgroup analyses.

*Limitations of the evidence included*. This study highlighted the limited reporting and consideration of equity in intervention effects. As few reviews used PROGRESS-Plus or completed a subgroup analysis, there were limited opportunities for comparisons within groups (e.g., by health-determinant category).

*Limitations to the processes*. The single Cochrane data source did not represent the breadth of evidence syntheses. It was not feasible to complete screening and data extraction processes in duplicate for all reviews, and formal quality appraisal was not conducted, a common challenge in methodological reviews [24,28,52]. It was also beyond the scope and resources of this study to consider the full range of approaches to equity assessment [24]. Focusing on use of PROGRESS-Plus, vulnerable populations and subgroup analysis was a feasible approach.

### 4.3. Implications

A more consistent use of tools and processes (PROGRESS-Plus, PRISMA-E) and transparency about which equity indicators were/were not used and why would improve the standards in reporting and subsequent understanding of health inequity. This responsibility falls on both reviewers and primary study authors, for whom PROGRESS-Plus provides a useful framework, in addition to guidance from numerous related articles and reviews published in the last 1–2 years [19,22,23,24,53,55,60,61]. In particular, primary studies in population health should engage with the relevant guidance for collecting and reporting equity data (e.g., [2,60,61,62]), which would give those synthesising evidence options for analysis by population subgroups. Standardisation in approach is not appropriate given the diversity in topics, populations and context. Yet, failure to improve practice to recognise specific groups, as observed with the historical under-representation of women or aggregation of minority ethnic groups in research [63,64], perpetuates invisibility and exclusion from the evidence, leading to evidence-based practice and policy not informed by their interests [65].

There is a need for greater consensus on socio-economic proxies (occupation, education, income, disadvantage measures) to facilitate meta-analyses and subgroup analyses. This is necessary for progress towards governments’ socio-economic equity goals [66,67,68] as reflected in the UN SDGs [to eradicate poverty (#1); achieve good health and well-being for all (#3); reduce inequalities (#10)] [69].

## 5. Conclusions

This systematic evaluation of population health evidence syntheses confirmed deficits in evidence through insufficient and inconsistent practice in reviews and primary studies, which ultimately limit evidence-based public health to address health inequity. Many reviews did not have an explicit equity focus and were often limited to the consideration of age, gender or sex and place. Few were able to complete a subgroup analysis to examine differential health outcomes/intervention effects, often prevented by a small number of primary studies or a lack of detail therein. This highlights missed opportunities for learning about health equity impacts, which could be improved through the routine use of PROGRESS-Plus for data extraction and equity consideration, with transparency regarding which indicators were/not reported and why.

## Figures and Tables

**Figure 3 ijerph-22-00851-f003:**
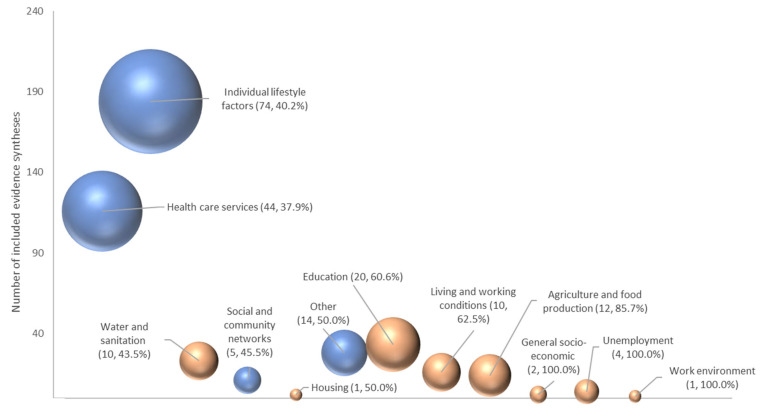
Bubble plot showing the number of reviews in each Dahlgren and Whitehead health-determinant categories (primary or secondary categories; vertical axis) and that planned subgroup analyses (size of bubble and figure in parentheses, percentage as proportion of total in category). Wider health-determinant categories shaded brown; other category types are shaded blue.

**Figure 4 ijerph-22-00851-f004:**
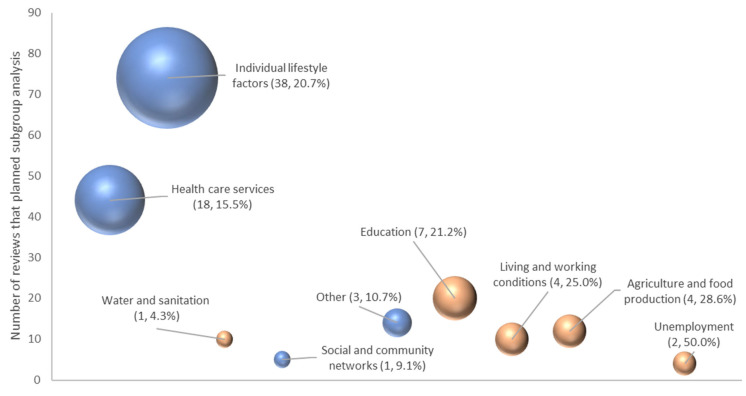
Bubble plot showing the number of reviews that planned (vertical axis) and completed (size of bubble and figure in parentheses, percentage as a proportion of total in category) subgroup analyses by Dahlgren and Whitehead health-determinant categories (primary or secondary categories). Wider health-determinant categories shaded **brown**; other category types are shaded **blue**. Note: housing, general socio-economic and work environment are not represented as no subgroup analyses were completed.

**Table 1 ijerph-22-00851-t001:** Number and percentage of reviews targeting different vulnerable population groups.

Vulnerable Population	Number of Reviews	Percentage of Reviews of Vulnerable Populations
Caregivers	4	14.3
Ethnic minorities	1	3.6
Experience of abuse ^a^	6	21.4
Families with attachment issues	1	3.6
Homeless	2	7.1
Lone parents with social welfare support	1	3.6
Mixed ^b^	1	3.6
Poor literacy	1	3.6
Refugees/asylum seekers	2	7.1
Substandard housing	2	7.1
Unemployed adults	1	3.6
Workers exposed to risk	3	10.7
Young offenders	2	7.1
Trachoma endemic area	1	3.6
Total	28	7.7

^a^ Includes review of women, children, partner violence or training of professionals regarding supporting/reporting of violence; ^b^ review authors defined vulnerable populations as groups not covered by health insurance scheme, including children, elderly, women, low-income individuals, rural populations, racial or ethnic minorities, immigrants, informal sector workers and populations with disability or chronic diseases.

**Table 2 ijerph-22-00851-t002:** Summary of reviews that cited use of PROGRESS or PROGRESS-Plus checklist.

Study	Title	Health-Determinant Categories	Subgroup Analysis		Checklist	How Progress Was Used (in Addition to Data Extraction)
			Planned	Completed		
Baker 2016 [37]	Interventions for preventing abuse in the elderly	Other	Y	N	PROGRESS-Plus	Planned subgroup analysis by geographical regions, sociodemographic characteristics of the target population. Planned to explore if equity gradient was apparent, if there was increasing gap and decreasing effectiveness by advantaged/disadvantaged populations, but there were insufficient data. ‘Evidence of consideration to equity issues’ reported for each study, but little narrative discussion of equity.
Brown 2019 [34]	Interventions for preventing obesity in children	Individual lifestyle factors	Y	Y	PROGRESS	Completed subgroup analysis by age; within age groups, reported where primary studies had undertaken subgroup analysis by SES, migrant status, ethnicity and rural/urban setting. Reported how studies had targeted disadvantage, with subheading for ‘equity and disadvantage’ and narrative on effects by age, gender, ethnicity, migrant status and urban/rural settings.
Centeno 2019 [38]	Fortification of wheat and maize flour with folic acid for population health outcomes	Agriculture and food production	N	N	PROGRESS-Plus	Recorded if studies included strategies to address diversity or disadvantage. Narrative synthesis described intervention impact by sociodemographic characteristics. Did not report findings by equity indicators.
Chamberlain 2017 [39]	Psychosocial interventions for supporting women to stop smoking in pregnancy	Individual lifestyle factors	Y	Y	PROGRESS-Plus	Used PROGRESS-Plus criteria to categorise interventions provided for vulnerable populations, which might impact vulnerability. Completed subgroup analysis by country income (LMIC, HIC), race/ethnicity (African American, Hispanic) and SES (low/not low), with narrative synthesis for other indicators.
Coren 2016 [40]	Interventions for promoting reintegration and reducing harmful behaviour and lifestyles in street-connected children and young people	Other	Y	N	PROGRESS-Plus	Used PROGRESS-Plus checklist alongside logic model. Planned subgroup analysis by age, gender, location of studies and HIC/LMIC, but there were insufficient data. Narrative examination of equity-related issues in primary studies, focusing on ethnicity, SES, gender, sexual orientation and disability.
Das 2019 [41]	Food fortification with multiple micronutrients: impact on health outcomes in general population	Agriculture and food production	Y	N	PROGRESS-Plus	Planned subgroup analysis by country income (LMIC/HIC) and age, but insufficient information as equity-related variables, and analyses were often missing from the primary studies. Included some descriptive analysis of the PROGRESS-Plus factors reported that highlighted deficient reporting in primary studies.
Garcia-Casal 2018 [42]	Fortification of maize flour with iron for controlling anaemia and iron deficiency in populations	Agriculture and food production	Y	N	PROGRESS-Plus	Recorded whether studies included specific strategies to address diversity or disadvantage. Planned subgroup analysis by gender was not possible. Table of studies reported against each PROGRESS-Plus indicator, with narrative synthesis to describe intervention impact by sociodemographic characteristics.
Hombali 2019 [30]	Fortification of staple foods with vitamin A for vitamin A deficiency	Agriculture and food production	Y	Y	PROGRESS	Recorded whether studies included strategies to address diversity or disadvantage. Planned subgroup analysis by age and gender, completed only for age. Table of studies reported against each PROGRESS indicator (and ‘Plus’) with narrative synthesis to describe intervention impact by sociodemographic characteristics.
Husk 2016 [43]	Participation in environmental enhancement and conservation activities for health and well-being in adults: A review of quantitative and qualitative evidence	Individual lifestyle factors	Y	N	PROGRESS-Plus	Planned subgroup analysis to explore potential impacts by SES, but no studies reported SES. Narrative analysis reported where included studies had undertaken subgroup analysis, but little overall discussion of equity impact.
Lhachimi 2020 [31]	Taxation of the fat content of foods for reducing their consumption and preventing obesity or other adverse health outcomes	Agriculture and food production	Y	N	PROGRESS	Planned subgroup analysis by country income, group income and age (children/adult), but insufficient data. Noted the ‘equity considerations’ for included studies, but there was no narrative synthesis by these factors nor discussion of equity.
MacArthur 2018 [51]	Individual-, family-, and school-level interventions targeting multiple risk behaviours in young people	Individual lifestyle factors	Y	N	PROGRESS-Plus	Planned subgroup analyses for all PROGRESS indicators (but did not specify variables for most). Data within each subgroup for outcomes were insufficient to complete these analyses. Included ‘equity’ section in the Results that provided a narrative description, and noted the limited information on which to draw inferences around equity.
Marx 2017 [35]	Later school start times for supporting the education, health and well-being of high school students	Education	Y	N	PROGRESS	Planned subgroup analyses by gender, age and/or grade, indicators of socioeconomic status and ethnicity, but there were too few studies. Authors included a brief narrative ‘report on equity’ in the Discussion.
McLaren 2016 [36]	Population-level interventions in government jurisdictions for dietary sodium reduction	Agriculture and food production	Y	Y	PROGRESS	Planned subgroup analysis to examine differential impact by multiple axes of social inequality based on PROGRESS indicators, but data only permitted this for gender/sex. Presented narrative synthesis to summarise for the remainder.
Morgan 2020 [45]	Caregiver involvement in interventions for improving children’s dietary intake and physical activity behaviours	Individual lifestyle factors	Y	N	PROGRESS-Plus	Extracted data for all PROGRESS-Plus indicators (including disability, sexual orientation, caregiver civil status). Planned subgroup analysis, but there were insufficient data. Discussion included consideration of implications for health equity and the research needs relevant to the promotion of health equity.
Mosdol 2017 [46]	Targeted mass media interventions promoting healthy behaviours to reduce risk of non-communicable diseases in adult, ethnic minorities	Individual lifestyle factors	N	N	PROGRESS-Plus	No planned subgroup analysis related to PROGRESS-Plus indicators. Narrative consideration of some equity issues (particularly by ethnicity).
Pena-Rosas 2019 [47]	Fortification of rice with vitamins and minerals for addressing micronutrient malnutrition	Agriculture and food production	Y	Y	PROGRESS-Plus	Planned and completed subgroup analysis by malaria endemic/malaria-free location (‘Place’). Included a table of studies reporting against each PROGRESS-Plus indicator and narrative synthesis to describe intervention impact by sociodemographic characteristics.
Pega 2013 [32]	In-work tax credits for families and their impact on health status in adults	General socio-economic	Y	N	PROGRESS	Included data on gender identity (and sexual orientation) and extracted data on inclusion of strategies for supporting disadvantaged populations. Planned subgroup analyses by ethnicity, family type (one-parent family, two-parent family), gender and income were not possible due to a small number of studies. Included an ‘Impact on equity’ section with narrative synthesis. Noted a lack of information available for subgroup analysis.
Petkovic 2021 [48]	Behavioural interventions delivered through interactive social media for health behaviour change, health outcomes and health equity in the adult population	Individual lifestyle factors	Y	N	PROGRESS-Plus	Planned harvest plots to assess the presence of gradients in effects across sex, ethnicity, SES and other PROGRESS-Plus characteristics for each outcome, but there were insufficient data. Used narrative synthesis with an ‘equity’ section in results, summarising data from four studies for which data were available.
Pfindern 2020 [33]	Taxation of unprocessed sugar or sugar-added foods for reducing their consumption and preventing obesity or other adverse health outcomes	Agriculture and food production; individual lifestyle factors	Y	N	PROGRESS	Planned subgroup analyses with data on most PROGRESS categories were not possible due to the inclusion of only one study (which also limited potential narrative synthesis around equity).
Shah 2016 [49]	Fortification of staple foods with zinc for improving zinc status and other health outcomes in the general population	Agriculture and food production	Y	Y	PROGRESS-Plus	Completed subgroup analysis by age. Table of studies reported against each PROGRESS-Plus indicator, and narrative synthesis describe intervention impact by sociodemographic characteristics (mainly limited to age groups).
von Philipsborn 2019 [50]	Environmental interventions to reduce the consumption of sugar-sweetened beverages and their effects on health	Living and working conditions; individual lifestyle factors	Y	Y	PROGRESS-Plus	Completed subgroup analysis by gender/sex. Narrative synthesis of studies reporting subgroup analyses by indicators of social disadvantage (‘SES’) and gender/sex and presented in a separate Appendix.

**Table 3 ijerph-22-00851-t003:** Number and proportion of reviews that planned or completed subgroup analysis using equity indicators.

PROGRESS-Plus Indicator Type	Planned		Complete		
	n	% ^a^	n	% ^a^	% ^b^
**P**lace	47	28.5	13	7.9	21.0
**R**ace/ethnicity	31	18.8	8	4.8	12.9
**O**ccupation	9	5.5	1	0.6	1.6
**G**ender/sex	69	41.8	24	14.5	38.7
**R**eligion	3	1.8	0	0.0	0.0
**E**ducation	11	6.7	0	0.0	0.0
**S**ocio-economic status (SES)	37	22.4	8	4.2	11.3
**S**ocial capital	2	1.2	0	0.0	0.0
**Plus** Personal characteristics (total)	91	55.2	30	18.2	48.4
—Variable: Age	91	55.2	30	18.2	48.4
—Variable: Disability	3	1.8	1	0.6	1.6
—Variable: Sexual orientation	1	0.6	0	0.0	0.0
Relationships (total)	2	1.2	0	0.0	0.0
—Variable: Family type	1	0.6	0	0.0	0.0
—Variable: Parents of child with disability	1	0.6	0	0.0	0.0
Time-dependant relationships (total)	3	1.8	0	0.0	0.0
—Variable: Residential history (sheltered/unsheltered housing)	1	0.6	0	0.0	0.0
—Variable: Exposure to injury risk	1	0.6	0	0.0	0.0
—Variable: Victim, perpetrator	1	0.6	0	0.0	0.0
Total count of indicator use	400		115		

^a^ % figures are proportions of 165 reviews that planned subgroup analyses using equity indicators. ^b^ % figures are proportions of 63 reviews that completed subgroup analyses using equity indicators.

## Data Availability

Data supporting reported results can be requested from the authors.

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
