# Peer review of "Systematic Evaluation of How Indicators of Inequity and Disadvantage Are Measured and Reported in Population Health Evidence Syntheses"

_ijerph, 2025, doi:10.3390/ijerph22060851_

Round 1
Reviewer 1 Report
Comments and Suggestions for Authors
The manuscript presents a systematic evaluation of population health reviews from the Cochrane Database (January 2013–February 2023) to assess how indicators of inequity or disadvantage are considered and reported in population health evidence syntheses.
The consideration of health inequality and disparities in population health is a highly relevant topic, and I found the findings from over 300 Cochrane reviews to be substantial and informative. However, I have a few concerns as outlined below:
-
While the authors have acknowledged the use of a single database as a limitation, the last search date being over two years ago raises concerns about the potential omission of relevant recent articles, which may affect the relevance and timeliness of the findings.
-
The exclusion of reviews focusing solely on LMICs may have limited the study’s ability to explore how inequality and disadvantage are addressed in different health system contexts. Including such reviews could have allowed for a comparison of the indicators used across global versus LMIC-only settings. Given the more apparent constraints in health systems, knowledge, and resources in LMICs, their inclusion may have yielded important insights.
-
To improve readability: In Figure 2, the number of reviews amounts to over 400, which exceeds the total number of included studies. I assume this is due to studies being categorized under more than one health determinant. I had to refer to the supplementary file to understand this. It would be helpful if the authors presented these data using infographics or bubble plots. This also applies to Supplementary Table 5—navigating 365 rows to understand population types and themes of vulnerability is difficult and could be made more accessible through a visual summary such as a bubble plot or similar format.
-
The description of excluded studies in Figure 1 and the supplementary files should be consistent. Additionally, some scoping reviews may contain data with subgroup or ad hoc analyses. Please verify whether reviews were excluded based on this criterion and consider including them if relevant analyses were present. A minor point: the citation format in Supplementary Files 2 and 5 differs from the main manuscript—please ensure consistency.
-
Given the lengthy review process, many high-quality reviews are now being published in non-Cochrane journals such as BMC Medicine. Therefore, I am not sure that focusing solely on Cochrane reviews can be considered a strength.
-
Please check for consistency in data presentation throughout the manuscript. For example, the abstract mentions “…a predominance related to individual lifestyle factors (n=155, 44.3%)…”, but I believe this should be (n=155, 42.7%)?
Reviewer 2 Report
Comments and Suggestions for Authors
Thank you for the opportunity to review “Systematic evaluation of how indicators of inequity and disadvantage are measured and reported in population health evidence syntheses.” Clearly the authors put in a high level of effort to abstract these papers and summarize the findings. I feel that the impact of this paper can be improved with some clarifications.
First, as the authors point out, some of the review authors may not be viewing their reviews as “population health,” thus, does the fact that this paper finds so few papers include the PROGRESS-plus framework variables actually evidence of lack of equity focus? More of a dive into what these original papers/reviews are used for, and the findings of the current paper is warranted. It could clarify the impact of the current paper.
For instance, almost half of all reviews focused in individual lifestyle behaviors. So, these papers, by default, may not engage in PROGRESS-related measures. But…maybe they should!
Why use the PROGRESS-plus framework? Are there others that could have been used? Why this one (and it’s inclusive list of variables)? More background work and motivation would be useful.
I like Figure 2, and I like the message that routine use of some framework (say PROGRESS-plus) could facilitate sub-group comparison.
Why exclude LMIC only reviews?
Some minor edits: At times “table” is used to refer readers, as opposed to “table #”.
Some of the text results skip to previous tables. Consider ordering the reporting of results to align with presentation in tables.
Some of the references were not formatted correctly in the draft.
